# Transforming specialized mental health practice: Insights from clients' perspectives on how to support recovery

Karin Lorenz-Artz[1,2]*, Joyce Bierbooms[1], Inge Bongers[1]

1 Tranzo, Tilburg School of Social and Behavioral Sciences, Tilburg University, Tilburg, The Netherlands,
2 Mental Health Care Institute Eindhoven, Eindhoven, The Netherlands

* c.a.g.lorenz@tilburguniversity.edu

## Abstract

To better address clients' needs, specialized mental health care (MHC) is changing to recovery-oriented, person-centered network care. This study was part of a broader Peer-supported Open Dialogue (POD) project and explores the experiences of clients living with severe mental illness (SMI) in a POD practice. The objective is to deepen the understanding of clients' lived experiences with POD to identify the specific behaviors, skills, and organizational elements they consider beneficial to their recovery. These insights offer practical guidance within a transformative care context, fostering the transformation at the personal level of MHC professionals committed to recovery-oriented, person-centered network care. Ultimately, this study aims to contribute to the development of new approaches for supporting the recovery of clients living with SMI. We conducted semi-structured interviews with 13 clients with SMI and one relative. In addition, we included data from five conversations held with clients with SMI within the POD project, two of whom were also interviewed in this study, as well as one relative. We used a hybrid analysis approach that combined deductive and inductive thematic analyses. We found that, at its core, the role of MHC professionals in clients' recovery revolves around strengthening three interrelated building blocks: promoting self-determination, creating and strengthening human connection, and establishing and facilitating reciprocal need- and ability-adapted collaboration. Underlying aspects of the building blocks revealed necessary competencies and supportive organizational elements, highlighting the complexities of power dynamics. Based on clients' experiences within a transforming specialized MHC practice, this study sheds light on the attitudes and behaviors of specialized MHC professionals that clients identify as supportive of recovery. Rather than fixed roles, the guiding role of MHC professionals in the recovery process requires a dynamic balance. This involves active engagement, navigating power dynamics and aligning with the needs and capabilities of everyone involved in the treatment.

**Data availability statement:** The datasets presented in this article are not readily available because the raw data supporting the conclusions of this article cannot be made completely anonymous. Requests to access the datasets should be directed to the institutional review board of MHC Eindhoven and Kempen (GGzE), wetenschapscommissie@ggze.nl.

**Funding:** The authors received no specific funding for this work.

**Competing interests:** The authors have declared that no competing interests exist.

**Abbreviations:** POD, Peer-supported Open Dialogue; OD, Open Dialogue; MHC, Mental health care; FACT, Flexible Assertive Community Treatment.

# 1 Introduction

Mental health care (MHC) is transitioning globally toward community-based, recovery-oriented, person-centered services [1–8]. Ethically, this approach is compelling, as no professional intentionally aims for compassionless or ineffective care [9]. While many MHC professionals claim to offer individualized, respectful care, practice often falls short [1,9–12]. Implementing recovery-oriented care remains challenging, as it requires not only systemic reform but also personal transformation among professionals [1,13–17].

This transformation entails a shift in values: recognizing individuals with severe mental illness (SMI) as whole persons, not merely clients [18–20], and enabling them to direct their own lives rather than being passive recipients of care [21]. It calls for a holistic approach beyond the biopsychosocial model [5,19], viewing recovery as a personal, evolving process that restores agency and meaning [14,22]. Rather than focusing solely on diagnosis and outcomes (clinical recovery), professionals must help clients pursue lives aligned with their own aspirations [9,23], empowering both them and their social networks [2,24–26]. The focus is on personal recovery, which aims not at normalization, but at enabling each individual to flourish as the unique human being they are [27].

However, this shift presents challenges. Recovery is multidimensional and marked by dualisms - clinical vs. personal, individual vs. social, process vs. outcome - that complicate practice [28–30]. For instance, balancing clinical stabilization with personal autonomy can be difficult. Similarly, while diagnosis may offer initial relief, it risks becoming a limiting identity, contributing to self-stigma and diminished hope [31,32].

## Peer-supported open dialogue

In the Netherlands, several specialized MHC organizations have explored and (partly) adopted Peer-supported Open Dialogue (POD) as a transformative model for recovery-oriented, person-centered network care. POD integrates peer workers - professionals with lived experience - and emphasizes transparency and shared decision-making, echoing the recovery movement's ethos of "nothing about us, without us" [33–35]. POD and OD are used interchangeably here, depending on peer involvement, but we refer to the same approach.

OD is recognized as a driver of systemic transformation in MHC [3,4,36,37], and endorsed by the WHO as a model of innovative practice [4]. OD exemplifies how professionals can navigate recovery dualisms and support meaningful change. It rejects biologically reductionist views, aims to reduce therapeutic hierarchies, and centers clients within relational systems [15,38]. Its collaborative, dialogical approach fosters responsive, continuous care and values lived experience [15,39,40].

Central to the approach are facilitated network sessions, led by at least two professionals trained in OD, a format that differs from conventional individual consultations. During these sessions, professionals may engage in "reflection moments," in which they briefly share their observations, thoughts, or emotional responses with each other in the presence of the client and their network. These reflections are offered

transparently and tentatively, fostering dialogue rather than imposing interpretations [41,42]. In line with the principle "nothing about me without me," no separate case discussions take place without the client present; decisions and clinical deliberations occur within the network meeting itself. OD does not preclude any form of treatment from the outset, nor does it operate with a predetermined agenda or predefined treatment goals, unlike care pathways organized around diagnosis-driven treatment planning. Instead, professionals follow the network's lead through open-ended questions and responsive engagement, allowing concerns, meanings, and potential directions to emerge within the dialogue. Any (evidence-based) intervention can be integrated when the network identifies it as helpful, based on shared understanding and collaborative decision-making. This approach seeks to empower individuals to take ownership of their own trajectories and actively shape their personal narratives [42]. Table 1 outlines the core OD principles.

Understanding this transformation requires deep engagement with client perspectives. Without their voices, recovery risks being defined by external stakeholders rather than those directly affected [7,43]. Research must intentionally include clients to ensure authenticity in recovery-oriented care development.

### Purpose

This study explores clients' experiences with POD to illuminate how recovery-oriented, person-centered care can be realized. By examining these lived experiences, it identifies specific behaviors, skills, and organizational conditions that clients find supportive. These insights may offer actionable guidance for professionals seeking to embody transformative care.

## 2 Method

### Design

This qualitative, practice-oriented field study was conducted from November 2018 to February 2021 within a broader POD project that established a multidisciplinary outpatient POD team (see Setting). Primary data collection for the present study consisted of semi-structured in-depth interviews exploring clients' experiences with POD care. In addition, existing qualitative material from project-based semi-structured conversations was included. These conversations had been conducted prior to the study period as part of routine quality improvement activities within the POD project and followed the same semi-structured format and topic list as the research interviews. Including this material allowed us to enrich the dataset while maintaining methodological consistency across data sources. All data were analyzed using a hybrid approach

**Table 1. Core principles of OD approach [40].**

| Principle | Short Description |
|---|---|
| 1. Immediate help | First meetings are arranged within 24 hours of initial contact to address crises promptly. |
| 2. Social network perspective | Engages the client's family, friends, and other key figures in network meetings, forming the backbone of treatment. |
| 3. Flexibility and mobility | Services adapt to the evolving needs of the client and network, including location and timing. |
| 4. Responsibility | The team that initiates care maintains responsibility throughout the treatment process. |
| 5. Psychological continuity | The same team remains involved over time, fostering trust and therapeutic consistency. |
| 6. Tolerance of uncertainty | Avoids premature conclusions, allowing space for open-ended exploration and shared meaning-making. |
| 7. Dialogism (polyphony) | Emphasizes genuine dialogue where all voices are heard equally, fostering mutual understanding. |

combining deductive and inductive thematic analysis [44,45]. This method integrates prior findings while revealing new insights to advance understanding [46].

## Ethics

The study received ethical approval from the Dutch Ethical Review Board of Tilburg University (REF EC-2018.91) and the institutional review board of GGzE (file ILB/2020021), which assessed both feasibility and ethics. Participants received an informational letter, were informed in advance that no compensation would be offered, and provided informed consent for the scientific use of their data.

## Setting

The present study was conducted within the Dutch specialized mental health care (MHC) system. In recent years, this system has increasingly emphasized recovery-oriented, network-based, and person-centered care [36,47]. Dutch society is relatively egalitarian, characterized by low power distance and a strong emphasis on consultation, collaboration, and individual autonomy [48]. These cultural and systemic features closely align with key principles of Open Dialogue, including shared decision-making, dialogical practice, and respect for multiple perspectives.

The study took place at GGz Eindhoven and the Kempen (GGzE), a large specialized MHC organization in the southern Netherlands that provides care to individuals with complex or acute mental health conditions and supports over 11,000 clients annually. GGzE operates in the Eindhoven-De Kempen region, which comprises 15 municipalities with approximately 600,000 inhabitants and includes both urban and rural areas. Eindhoven, the largest city in the region, is a medium-sized urban center with a strong focus on technology, design, and innovation [49]. While the region overall has an average socioeconomic status (SES-WOA score 0.04), the neighborhoods in which most participants resided have slightly lower socioeconomic status than the national average (SES-WOA −0.27) [50].

Within GGzE, ambulatory care for people with severe mental illness (SMI) was largely organized through Flexible Assertive Community Treatment (FACT) teams, which employ a multidisciplinary outreach model for individuals who may not benefit from conventional outpatient services [51]. Following completion of the UK-based postgraduate training "Peer-supported Open Dialogue, Social Network and Relationship Skills", one FACT team fully transitioned to POD practice. The resulting POD team was multidisciplinary, consisting of POD-trained psychiatrists, psychologists, social workers, occupational therapists, and peer-support workers (paid professionals with lived experience).

The team serves approximately 285 clients, with 20–30 new referrals annually, and operates in accordance with the OD approach, as outlined in the Introduction and summarized in Table 1. Previous studies have described the implementation of POD within this team and its associated challenges [17,52]. The team provides a continuous, person-centered treatment pathway, covering the clinical process from initial contact to completion of care. Network meetings form the core of the approach, with additional interventions and support offered flexibly in response to clients' needs and preferences. Client transitions from POD follow standard mental health practice: when network discussions indicate that professional support is no longer needed, POD staff gradually step back at a pace tailored to the client's needs.

## Recruitment and consent

During the POD project, clients receiving or entering POD care could apply to participate. Recruitment occurred in two phases (Nov 2018–Mar 2019; Sep 2020–Feb 2021), with treatment teams actively involved. Eligible participants were adults with SMI, experiencing significant social and societal limitations for at least one year, and who may not have benefited from traditional mental health services [51]. Inclusion was encouraged regardless of recovery stage, treatment motivation, their willingness to seek help, or attitude toward POD (positive, negative, or neutral). Clients with intellectual disabilities, memory impairments, or in psychological crisis were excluded. Capacity to provide informed consent was

assessed by the principal clinician, who made a professional judgment on client's ability to consent; only those deemed capable were approached, in accordance with the consent procedure approved by the ethics committee.

MHC professionals identified suitable, willing clients, who were then sent an information letter and asked to respond within two weeks. Clients could invite network members to participate. Upon signing informed consent, interviews were scheduled at a location chosen by the client, provided it was quiet and suitable. All communication with network members was facilitated through the client.

## Participants

Clients eligible for the present study were adults with SMI and significant functional limitations persisting for at least one year, requiring coordinated and long-term mental health care.

All participants were receiving care within a FACT service and were transitioned to a POD team at the start of the POD pilot, with the exception of two participants who entered POD care at a later stage. As part of this transition, all clients received POD-based care and were informed about the dialogical and network-oriented approach. Each client is supported by two POD professionals who facilitate network meetings - preferably at home - on weekdays. Care intensity is adjusted as needed, with crisis support available when required.

We interviewed 13 clients (7 male, 6 female) and one female relative. Additionally, data from five project-based conversations were included, involving five clients (two also interviewed) and one female relative. In total, data from 16 clients (9 men, 7 women) and two female relatives were included in the analysis. Ages ranged from 19 to 65, with a mean of 47.57 years (SD = 10.25, excluding the 19-year-old). Prior to participation in the study, six clients had taken part in more than 50 network meetings, four in more than 30 meetings, and six in more than 20 meetings, indicating substantial exposure to the POD approach. All clients experienced long-term SMI with functional impairments. Mental health conditions, as self-reported by participants, included psychotic disorders, trauma-related difficulties, anxiety, depression, bipolar disorder, and addiction.

## Interview procedures

Interviews were conducted in Dutch and lasted 1–1.5 hours. Four interviews took place in clients' homes; the remainder were held on-site at the MHC organization. One interview included a client's mother; all others were individual. Interviews were guided by a semi-structured topic list based on POD principles and recovery literature [41,53]. Open-ended prompts allowed participants to elaborate on topics most relevant to their experiences. Core themes included involvement, support, struggles, needs, flexibility, network meetings, reflection, and recovery. Representative example questions included: "How did you experience the support from the POD team in your recovery process?", "In what ways were you involved in decisions about your care?", and "Were there aspects that did not contribute to your recovery?" The topic list was deliberately flexible, allowing participants to discuss both positive and challenging experiences and to focus on issues most meaningful to them.

## Data management and analysis

In addition to the interviews, existing data from five project-based semi-structured conversations was included, using the same semi-structured format and topic list as the research interviews. All interviews and project-based conversations were audio-recorded and transcribed verbatim and analyzed using thematic coding in Atlas.ti [44], following Braun and Clarke's six-phase framework [44] through an iterative, hybrid approach [45]. Following a phenomenologically informed approach [46] focusing on participants' experiences, initial codes were generated using both deductive and inductive techniques, informed by POD principles, recovery terminology, and study objectives (e.g., reflection moments, responses to problems, tolerance of uncertainty, team meetings, network participation, connectedness, client narratives, support, challenges, and prerequisites). New codes were added as needed.

Codes were then clustered into latent subthemes through reflective iteration, which were organized under broader themes capturing underlying assumptions and conceptualizations beyond explicit statements [54], reflecting an interpretive, hermeneutic approach [55]. Themes were reviewed, refined, and defined. Analyses were conducted on the original Dutch transcripts, and the results were subsequently translated into English with co-author review to ensure conceptual accuracy.

### Reflexivity

To account for the influence of researchers' professional backgrounds and analytic positioning on data collection and interpretation, reflexivity was explicitly addressed throughout the study.

Two researchers conducted the interviews. The first round was led by a POD-trained researcher with clinical experience in crisis care for individuals with SMI; the second by an MHC nursing trainee with extensive experience but no POD training. Neither had a treatment relationship with participants. Their professional backgrounds were considered instrumental in eliciting rich, reflective data.

Regarding the analyses, to preserve participants' perspectives, we deliberately avoided retrofitting themes into predefined frameworks like POD principles [41]. Coding uncertainties were resolved collaboratively, and preliminary findings were shared with the POD team to validate interpretive credibility [56]. The POD team recognized the preliminary findings as consistent with their experience with the clients, confirming that the analyses accurately reflected participants' perspectives and providing no additional comments.

## 3 Results

Three interrelated building blocks emerged as central to POD's contribution to clients' recovery: (3.1) promoting self-determination, (3.2) creating and strengthening human connection, and (3.3) establishing and facilitating reciprocal, need- and ability-adapted collaboration. Together, these building blocks formed the foundation through which participants experienced the role of POD care in their recovery processes.

Across interviews, most participants described POD care as empowering. This was reflected in shifts such as renewed hope, strengthened self-assurance, greater openness to vulnerability, improved self-awareness, restored relationships, and enhanced coping in daily life. For some, this empowerment was experienced as a renewed sense that change remained possible, even after long periods of stagnation, as one participant reflected: *"I always had the feeling like, I'm not learning anything anymore. And now, that something is being set in motion […] that change is still possible." (Participant 15).* Several participants emphasized active engagement in decision-making, developing inner resilience over time, being transparent with professionals about vulnerabilities, and taking personal responsibility.

At the same time, recovery trajectories differed. While some participants described broader and more sustained changes, others experienced the impact of POD as more limited in scope and primarily confined to the network meetings themselves. As one participant noted the impact of the network meetings as follows: *"The conversations give me a sense of self-worth […] I can vent […] then I am relieved for a moment" (Participant 1).*

Returning to the three building blocks outlined above, Table 2 provides an overview of the corresponding subthemes, followed by detailed elaboration in the subsections below.

### Promoting self-determination

A central way in which empowerment became visible in participants' accounts was through the promotion of self-determination. This was reflected in four interrelated aspects: (1) instilling self-confidence, (2) providing space to make one's own choices, (3) aligning with one's own story, and (4) Balancing tolerance and the need for guidance.

PLOS Mental Health

**Table 2. Overview of the three building blocks, their subthemes, and concise descriptions.**

| Building block | Subthemes | Description |
|---|---|---|
| **Self-determination** | *Instilling self-confidence* | Participants valued being recognized as capable; professionals emphasized strengths and offered positive reinforcement, enhancing motivation and self-worth. Some participants, however, did not experience this as impactful. |
| | *Providing space to make one's own choices* | Participants valued autonomy in care decisions, with professionals respecting choices and encouraging independence. The safe, supportive environment allowed participants to make decisions, solve problems, and see themselves as capable. |
| | *Aligning with one's own story* | Participants appreciated professionals' deep engagement with their personal stories, providing freedom to interpret experiences and fostering trust. Peer workers' lived experience and the presence of two professionals enhanced understanding, objectivity, and emotional depth. |
| | *Balancing tolerance and the need for guidance* | Participants valued the balance between autonomy and professional guidance. Allowing self-direction was empowering but could be challenging; readiness and clear support were key to prevent feeling overwhelmed. |
| **Human connection** | *Genuine Involvement* | Participants experienced professionals as emotionally engaged, nonjudgmental, and deeply committed. Frequent contact, availability, and attention to everyday matters fostered trust, safety, and a sense of being valued. |
| | *A more egalitarian personal connection* | Participants valued egalitarian relationships, feeling respected as humans rather than patients. Professionals' openness fostered emotional reciprocity while maintaining clear professional boundaries. |
| | *Normalizing experiences: a person-centered approach* | Participants felt recognized as whole persons beyond diagnoses. POD's person-centered approach reduced stigma and supported recovery. |
| | *Nothing about me without me* | The reflection moments fostered openness, respect, and trust. Participants felt empowered and included, while also accepting that some discussions outside their presence could enhance care. |
| **Reciprocal collaboration** | *Collaboration on an equal footing* | Participants collaborated side by side with professionals in their recovery, actively sharing decisions and responsibilities. |
| | *Responsive care: flexibility, availability and support* | Participants experienced flexible, client-paced care, with professionals readily available through home visits, phone contact, and timely appointments. Emotional attunement and calm, unhurried sessions fostered trust, support, and empowerment. |
| | *Holistic collaboration: integrating social- and professional networks* | Participants engaged in holistic collaboration, integrating personal and professional networks. POD facilitated involvement of social networks, addressed relational dynamics, and adapted support across recovery phases, providing tailored, multidisciplinary care. |

**3.1.1 Instilling self-confidence.** All participants valued being seen as capable individuals rather than passive patients. According to participants, POD professionals explicitly highlighted progress, reflected back personal qualities during meetings, and discussed strengths in reflection moments. These moments were frequently described as particularly valuable, as they enhanced motivation and strengthened self-worth. One participant described how positive acknowledgment during meetings rekindled a sense of personal worth and motivation:

*"Involvement, openness, honesty. For me, as an addict who has lost all of these things […] to get them back, that is so important. That's much more important than the treatment method. It feels so good when people speak positively about someone. It makes you want to do your best." (Participant 4)*

However, not all participants experienced this emphasis on strengths as meaningful:

*"I don't care about compliments […] they do it from their work situation […] You go there for an hour, and that's it." (Participant 1)*

### 3.1.2 Providing space to make one's own choices.

All participants appreciated autonomy in shaping their care, deciding on meeting times, locations, invitees, and treatment goals. According to participants, POD professionals respected their decisions and encouraged independence, which contributed to a sense of agency rather than illness identity.

As one participant explained, being given choice altered how they positioned themselves within care:

*"The choice is left with the client. It gives you less of a feeling that you're really sick and more confidence in yourself."* *(Participant 16)*

Several participants further noted that POD professionals deliberately refrain from intervening when clients are able to manage situations themselves. This was experienced not as neglect, but as professionals being present while creating a safe atmosphere and space for reflection and shared exploration. Within this space, participants felt encouraged to make their own decisions, develop solutions, and articulate their own perspective, positioning themselves as experts in their own lives. For some, this non-directive and non-demanding stance fostered openness and courage in dealing with difficult experiences:

*"You have to find a way to make life bearable […] through POD, you deal with it a bit easier because you can be open about the tough things. […] And that's what POD does too, they expect nothing from you, I expect nothing from them, but they give me so much […] That gives me courage."* *(Participant 8)*

### 3.1.3 Aligning with one's own story.

This subtheme refers to the way POD professionals attuned themselves to clients' personal narratives, supporting them in remaining authors of their own stories. Rather than reinterpreting or reframing experiences from a professional standpoint, participants described how professionals created space for clients to articulate, explore, and make meaning of their difficulties in their own terms. This alignment was experienced as fostering trust, emotional depth, and a sense of narrative ownership within the meetings.

Participants described POD professionals as actively and attentively engaging with their personal stories. One participant emphasized this experience of being fully accompanied within their own story:

*"They're truly in your story […] Not on the sidelines […] Very involved."* *(Participant 12)*

Participants further explained that the presence of two professionals during network meetings strengthened this process. Having two perspectives was described as enhancing objectivity and reducing the risk of unilateral interpretations, thereby limiting the extent to which professionals might impose their own judgments on clients' narratives.

As one participant explained:

*"The fact that there are two people present […] brings some objectivity."* *(Participant 11)*

The relatively unstructured character of the meetings was also experienced as supportive of alignment with one's own story. Participants described how conversations could unfold organically, allowing previously unspoken or sensitive experiences to surface. This openness created space for vulnerability and disclosure beyond what some had encountered in more structured therapeutic settings. This is illustrated by a participant who reflected:

*"It's always very heavy […] I've shared things […] I never shared with other therapists."* *(Participant 8)*

Most participants also highlighted the role of peer workers within this process. Their experiential knowledge was experienced as enabling direct empathy and rapid recognition of core issues, which reduced the need for extensive explanation.

According to participants, this form of recognition supported alignment with their own lived experiences, rather than externally imposed interpretations. As one participant described:

> *"Because he (the peer worker) had such specific issues, the peer worker picks it out […] That gives him a lot of peace."* (Participant 10)

**3.1.4 Balancing tolerance and the need for guidance.** Participants described how POD professionals deliberately created space for self-direction and personal responsibility. In practice, this meant that professionals refrained from taking over decisions and allowed participants to take initiative in shaping next steps. For several participants, this tolerance was experienced as empowering, fostering pride and a sense of ownership over their recovery process. As one participant reflected:

> *"I'm very proud […] through my willpower and effort, I managed to achieve this."* (Participant 4)

At the same time, participants emphasized that this approach required adequate information, timing, and attunement. Autonomy was experienced as supportive only when participants felt sufficiently informed and emotionally ready to take the lead. When professionals expressed uncertainty at vulnerable moments, this could instead amplify distress. One participant described how a perceived lack of guidance intensified his struggle:

> *"Then you come with that help request and the professional says, 'I don't know'. […] That makes it all much worse."* (Participant 4)

Thus, balancing tolerance and guidance emerged as a dynamic process, in which professionals needed to carefully attune the degree of direction to participants' readiness and needs.

**Creating and strengthening human connection**

Under this second building block human connection refers to an explicitly reciprocal, person-to-person form of relating, in which both professionals and clients remain present as human beings rather than solely as role-bearers. Creating and strengthening human connection emerged as an important dimension of how participants experienced POD, reflected in four interrelated aspects: (1) genuine involvement, (2) a more egalitarian personal connection, (3) normalizing experiences: a person-centered approach, and (4) nothing about me without me.

**3.1.5 Genuine involvement.** Participants consistently described POD professionals as emotionally engaged, nonjudgmental, and deeply committed. The calm, open atmosphere and frequent face-to-face contact fostered trust and a sense of safety. Availability via mobile phone and attention to everyday matters further reinforced this sense of connection. According to participants, professionals demonstrated genuine emotional involvement not merely as experts, but as fellow human beings. This approach enhanced feelings of being valued and respected:

> *"It feels like family, like care with love. […] They gave me back my self-worth."* (Participant 4)

**3.1.6 A more egalitarian personal connection.** Participants valued the egalitarian dynamic of POD care, feeling respected and understood as human beings rather than as patients. They described how professionals' openness about their own experiences reinforced this sense of companionship, enriching interactions and fostering emotional reciprocity:

> *"I don't feel like a client, but like myself."* (Participant 8)

At the same time, participants recognized that professionalism remained crucial. Several participants acknowledged the distinction between professional roles and personal relationships. While they appreciated personal sharing, boundaries were maintained to distinguish the professional role from that of the social network. Some reflected critically on the limits of an egalitarian relationship:

*"Equal care… but only to a certain extent. […] They won't come to me unless they're getting paid. Otherwise, they stay home, right?" (Participant 1)*

Another participant emphasized the importance of balancing personal disclosure with professional responsibility:

*"They stress that they're also human and face challenges. Sharing personal things can add an extra dimension, […] but you have to know where the boundary lies." (Participant 6)*

Overall, these accounts illustrate that the more egalitarian connections in POD care are experienced as relationally close yet professionally grounded, supporting trust and authentic engagement while maintaining clear boundaries.

**3.1.7  Normalizing experiences: a person-centered approach.**  Participants consistently described POD care as normalizing, emphasizing a holistic, person-centered ethos, with support addressing broader aspects of daily life and not solely focused on treating mental health problems. They felt recognized beyond their diagnoses, which contributed to reduced stigma and strengthened recovery:

*"You're not your depression, your psychosis, or your negative thoughts. They're there, but you're not defined by them." (Participant 16)*

Some participants highlighted an additional aspect: when professionals treat clients as whole persons rather than as patients, this can also imply being held to everyday social norms. For example, punctuality was mentioned as part of being regarded as a whole person, implying accountability in ordinary interactions.

While both 3.2.2 and 3.2.3 reflect more egalitarian relationships, the former emphasizes emotional parity, while the latter highlights normalization as a therapeutic stance.

**3.1.8  Nothing about me without me.**  Reflection moments, in which POD professionals discussed topics or observations from the network meeting in the presence of clients and their network, were new to all participants and fostered openness, respect, and trust. Though initially unfamiliar, most participants found these moments clarifying and empowering. Participants noted that professionals adopted a humble stance, validating their experiences and strengthening the sense of connection. One participant highlighted the value of openly addressing issues or experiences that surfaced during the network meetings:

*"There are no taboos! […] Anything can be discussed." (Participant 16)*

While participants appreciated being included in discussions, many recognized that some conversations could occur in their absence. They trusted the team's integrity and expertise, and several noted that such discussions could, in certain cases, support their recovery by leveraging additional knowledge and experience:

*"I still think that when they discuss things among themselves, more knowledge and experience comes together." (Participant 1)*

**Establishing and facilitating reciprocal need- and ability-adapted collaboration**

Participants emphasized the importance of collaboration in which professionals work alongside clients, adapting to their needs and abilities. This facet of POD care encompasses balancing autonomy, responsiveness, and integration of both social and professional networks. Three interrelated aspects emerged in this third building block: (1) collaboration on an equal footing, (2) responsive care: flexibility, availability, and support, (3) holistic collaboration: integrating social- and professional networks.

3.1.9 **Collaboration on an equal footing.** Participants described working with POD professionals as a shared recovery process rather than a traditional "care for" approach. Professionals were experienced as working alongside clients, characterized by mutual respect and emotional reciprocity, while supporting autonomy and remaining engaged in decision-making. As one participant explained:

> "Not hand in hand, but side by side. […] The responsibility for my recovery is my own, but they contribute to that." (Participant 12)

This sense of equality was particularly evident in shared decision-making around treatment goals and interventions. Participants described being actively involved in choosing next steps, weighing options, and reflecting together on consequences. Being included in these discussions strengthened their sense of agency.

At times, this emphasis on autonomy also created tension within participants' personal networks. One participant described frustration when family members attempted to shield him from difficulties, whereas he preferred to deepen understanding for his situation. Although the meeting structure allowed everyone to speak, he felt his concerns were overshadowed by those of his relatives, illustrating that maintaining equality in collaboration requires continuous attention.

At the same time, maintaining equality did not mean that professional guidance was absent. Several participants acknowledged that collaborative care sometimes required professionals to take a more encouraging or activating role. For example, one participant suggested that group treatment participation could be more actively promoted, as its value often became apparent only in retrospect. Equal collaboration was therefore experienced as a dynamic balance between autonomy and professional input.

3.1.10 **Responsive care: flexibility, availability and support.** Participants emphasized the importance of flexibility and availability, especially during moments of increased distress. Because recovery was experienced as unpredictable, participants valued care that adapted to their changing needs rather than adhering to fixed schedules. This flexibility was reflected in practical arrangements, including adjusting meeting times or conducting home visits during crises. As one participant explained, home visits allowed professionals to engage at "the right moment":

> "When you're feeling so bad, the last thing you want is to go somewhere. […] When they come to your home, I think they really see you at the right moment." (Participant 10)

Participants emphasized that knowing they could contact professionals at any time - via phone or immediate appointments - provided emotional security and reduced anxiety, even when support was not immediately required. One participant noted:

> "I can always call. Just knowing that helps me feel calmer." (Participant 13)

Similarly, rapid access to multidisciplinary professionals was highly valued

> "If you need to see a psychologist, you can go this afternoon, not wait eight weeks." (Participant 10)

Participants described meetings as calm-paced, meaning that conversations proceeded at a relaxed, unhurried rhythm, providing space for intense emotions, reflection, and deep processing. When professionals extended sessions to accommodate participants' needs, as one participant noted, *"They extended the session when they saw I needed more time" (Participant 15)*, this was experienced as attentiveness and contributed to feeling genuinely heard.

In essence, participants experienced POD's responsive care - anchored in flexibility, accessibility, and emotional attunement - as central to feeling supported and empowered throughout their recovery.

**3.1.11 Holistic collaboration: integrating social- and professional networks.** Participants emphasized that collaboration extended beyond the professional team to include their social networks, integrating both family members and other significant others into the recovery process. They particularly valued POD's encouragement to actively involve personal networks.

In practice, participants reported that network meetings were used to address relational tensions, clarify misunderstandings, and reconsider family roles and responsibilities. Several participants described these meetings as opportunities to explore long-standing interactional patterns. One participant reflected on how such discussions enabled family members to better understand and support one another:

> *"We really address issues that make life heavier. We solve puzzles and misunderstandings within the family, and we can be open and honest with each other. This helps move the process forward, which is very important." (Participant 16)*

Through such exchanges, collaboration became a vehicle for restoring relationships and strengthening shared responsibility in the recovery process.

At the same time, participants acknowledged that involving social networks was not always straightforward. Some participants described emotional strain on family members, cultural taboos surrounding mental health, and experiences of being misunderstood or blamed. In these situations, the presence of POD professionals was experienced as protective and validating. One participant highlighted that when family support was limited, the involvement of professionals provided a sense of recognition and validation:

> *"It's lonely when no one sees what's really happening. […] My mother[…] sabotages me in a certain way. […] Others blame me for my problems. […] While professionals who have studied for this, they recognize it." (Participant 5)*

Here, professional involvement functioned as a counterbalance, offering recognition and interpretive support when social networks were unable or unwilling to do so.

Holistic collaboration also encompassed integration within the professional team. Participants consistently valued the team's multidisciplinary composition and the continuity of familiar professionals, which fostered trust, reduced the need to repeatedly retell personal histories, and created a sense of safety. At the same time, one participant reflected that these strong relational bonds could create a sense of loyalty, complicating the decision to seek support from another team member when needed.

Participants appreciated the team's adaptability across recovery phases. Shifts in focus - from peer support to employment coaching to case management - reflected evolving needs and reinforced a sense of personalized, responsive care.

## 4 Discussion

This study aims to contribute to the transformation of MHC toward recovery-oriented, person-centered network care by exploring clients' experiences with POD care. Recognized as a driver of this systemic change [3,4,36,37], POD offers insight into how recovery-oriented care can be realized in practice. From the client perspective, three interrelated building

blocks emerged as central to professional support: (1) promoting self-determination, (2) fostering human connection, and (3) facilitating reciprocal, need- and ability-adapted collaboration.

Rather than "caring for," clients emphasized the importance of "collaborating with" professionals. Key competencies included respect, trust, compassion, nonjudgmental listening, belief in recovery potential, alignment with personal narratives, openness, taking time and accessibility. These qualities align with the POD approach [42] and broader recovery literature [e.g., 53, 57].

Organizational elements that supported these competencies included network meetings with reflection moments involving at least two POD professionals [42], peer worker involvement, consistent teams, direct availability, and multidisciplinary access. While many of these features overlap with models like FACT [51], clients highlighted the importance of balancing consistency with flexibility. Strong bonds with caregivers, while valuable, could sometimes inhibit help-seeking from others, underscoring the need for open dialogue to address relational ruptures. Such moments, if explored constructively, may foster new ways of experiencing oneself and others [58].

Network meetings with reflection moments involving at least two POD professionals were seen as particularly innovative. In these meetings, professionals openly shared personal impressions and emotional responses, rather than maintaining the more neutral stance typical of conventional care. Conducted with dialogism, tolerance of uncertainty, and reflective practice [41,42,59,60], they created "free spaces" for clients to pursue meaningful, autonomous lives [34]. These moments also guided professionals in adopting empowering, equalizing roles while maintaining expertise [52], embodying the recovery movement's ethos of "Nothing about me, without me" [34]. Finally, the presence of two professionals and a peer worker helped prevent the imposition of personal judgments on clients' narratives and reduces the risk of professionals being influenced by their own preconceptions. This may enable MHC professionals to adopt a more guiding, non-impositional stance [23,42,61–63], reinforcing the collaborative nature of recovery-oriented care [12].

This study highlights the nuanced tension between empowerment and control, particularly within the building block of promoting self-determination. Literature advocates for professionals to adopt a guiding role, relinquishing control and recognizing clients as agents of their own recovery [23,42,61–63]. Our findings support this, emphasizing clients' desire to actively participate in decision-making [62]. However, clients also stressed the need for professionals to provide clear information and maintain appropriate oversight. Excessive responsibility placed on clients may undermine care quality and professional expertise [64], especially during crises where risk management and autonomy may conflict [65]. While some studies warn that empowerment can mask reduced support [66,67], this concern was not raised by our participants. Instead, they called for a dynamic balance in which control should be flexibly negotiated rather than entirely relinquished.

This balancing act also surfaced in network meetings, where professionals must navigate complex interpersonal dynamics. When one voice dominates [68] or, as in our study, when network concerns diverge from the client's, professionals may need to actively structure dialogue to ensure inclusivity [68]. This can challenge shared decision-making and the integrity of all three building blocks. POD offers strategies to manage this, such as turn-taking and reflective listening [41,59,60], allowing each narrative to be heard without steering the conversation [60]. In such cases, professional authority may be necessary to facilitate a more balanced inclusive dialogue, to empower clients and uphold collaborative principles.

While clients in this study experienced their relationships with professionals as more personal and egalitarian, they also recognized enduring structural asymmetries: professionals remained paid employees with formal responsibilities and institutional authority. The OD approach has been described as challenging established norms and power structures through a paradigm shift [69–71]. Our findings suggest, however, that this does not imply the disappearance of hierarchy, which aligns with the OD perspective [72]. Rather than eliminating hierarchy, POD appears to recalibrate it, shifting from directive control toward responsive guidance within an inherently asymmetrical relationship.

Another tension concerns availability. Clients valued the consistent presence of POD-trained professionals, not only in crises but also for emotional support. This raises questions about the role of specialized MHC professionals in

community-based care. While continuous availability may foster trust, it also challenges feasibility and the normalization goals of community care [10]. Clients noted that their social networks often struggled with the intensity of their needs, sometimes withdrawing altogether, a concern that was also found in literature [73]. Conversely, over-involvement or rigid helper roles among significant others may hinder autonomy and reciprocal relationships [74–77]. Thus, while network involvement is widely seen as beneficial, this study underscores the need for deeper understanding of the third building block - reciprocal collaboration - within complex interpersonal dynamics, while upholding the other two building blocks, which is in line with earlier findings [73].

Taken together, these findings illustrate how recovery-oriented, dialogical principles are enacted in practice, while highlighting tensions that warrant reflection. They should be interpreted in light of the Dutch socio-cultural and healthcare context, which is characterized by low power distance, a tradition of consultation, and individual autonomy [48]. These conditions may have facilitated clients' acceptance of collaborative practices within POD, so caution is warranted when considering transferability to settings with different structures or professional hierarchies.

## Strengths and limitations

This study offers rich qualitative insights into clients' experiences with recovery-oriented, person-centered network care within POD practice. A key strength lies in its grounding in real-world experiences and the researchers' expertise in engaging clients with severe and persistent mental illness, a population that is difficult to reach and underrepresented in the literature, providing detailed accounts that yield findings relevant to science, practice, and policy.

However, limitations must be acknowledged. Recruitment was mediated by POD professionals, potentially introducing selection bias despite efforts to ensure diversity. The participating POD team has a caseload of about 285 clients, and data from 16 clients (9 male, 7 female) and two female relatives were analyzed. The exact number of ineligible clients (e.g., with intellectual disability, memory impairment, acute psychological crisis, or inability to provide informed consent) is unknown. Although the participants represent a small proportion of the caseload, we consider it representative for the caseload, as the group was heterogeneous, encompassing a range of diagnoses and a variety of attitudes toward POD (positive, neutral, and critical).

We recognize that individuals who chose to participate may differ from those who declined; people less inclined to share their views might hold different perspectives, which could influence the findings. Individual differences in communication and familiarity with POD were not systematically accounted for, which may also have shaped the results. Moreover, the social network perspective -central to POD - was underrepresented, with only two interviews including network participants. Possible barriers included timing (during office hours), limited network involvement in the treatment, and client hesitancy. While more proactive support to involve network members might have increased participation, clients' autonomy was prioritized.

Despite these limitations, we observed that data saturation was reached during analysis. In addition, the anonymized, aggregated findings were presented to the treatment team, who confirmed their consistency with broader clinical experience and offered no additional feedback. Together, these observations support the credibility and transferability of the findings despite the small sample, which should be interpreted as transferable insights rather than statistically generalizable results.

## Conclusion and future research

This study deepens understanding of the attitudes and behaviors that clients perceive as supportive in recovery. The guiding role of MHC professionals centers on strengthening three interrelated building blocks: (1) promoting self-determination, (2) fostering human connection, and (3) facilitating reciprocal collaboration. This role requires dynamic engagement, balancing power, and adapting to individual needs. Organizational factors, such as structured reflection and consistent teams, also play a vital role.

Future research should intentionally broaden the participant base to include clients - particularly those less inclined to participate - as well as a wider range of professionals and social network members, to deepen understanding of how mental health professionals can optimally support recovery. Investigating how professionals navigate complex relational dynamics and tailor their approach to different individuals and contexts is essential. In particular, exploring the interpersonal processes within care networks will help refine network-based recovery care.

Additionally, while the therapeutic relationship is known to actively influence recovery [78], our findings suggest that self-determination and reciprocal collaboration may also function as independent contributors to recovery. Further research should examine their direct impact on recovery outcomes and how this varies across populations.

We hope these insights support the continued development of recovery-oriented, person-centered network care, especially for people living with SMI.

## Acknowledgments

We thank all participants for participating this study.

## Author contributions

**Conceptualization:** Karin Lorenz-Artz, Joyce Bierbooms, Inge Bongers.

**Data curation:** Karin Lorenz-Artz.

**Formal analysis:** Karin Lorenz-Artz.

**Methodology:** Karin Lorenz-Artz.

**Project administration:** Karin Lorenz-Artz.

**Supervision:** Joyce Bierbooms, Inge Bongers.

**Validation:** Joyce Bierbooms, Inge Bongers.

**Writing – original draft:** Karin Lorenz-Artz.

**Writing – review & editing:** Karin Lorenz-Artz, Joyce Bierbooms, Inge Bongers.

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
