## [Decision Letter · Decision Letter 0]

2 May 2025

PMEN-D-25-00075

Transforming specialized mental health practice: insights from clients’ perspectives on how to support recovery.

PLOS Mental Health

Dear Dr. Lorenz-Artz,

Thank you for submitting your manuscript to PLOS Mental Health. After careful consideration, we feel that it has merit but does not fully meet PLOS Mental Health’s publication criteria as it currently stands. Therefore, we invite you to submit a revised version of the manuscript that addresses the points raised during the review process.

We look forward to receiving your revised manuscript.

Kind regards,

Giovanni de Girolamo, M.D.

Academic Editor

PLOS Mental Health

Journal Requirements:

1. Please describe in your methods section how capacity to provide consent was determined for the participants in this study. Please also state whether your ethics committee or IRB approved this consent procedure. If you did not assess capacity to consent please briefly outline why this was not necessary in this case.

2. We ask that a manuscript source file is provided at Revision. Please upload your manuscript file as a .doc, .docx, .rtf or .tex.

Additional Editor Comments (if provided):

The paper needs a MASSIVE reduction in length, from over 10,000 words to not more than 5,000, possibly to 4,000 words. The text has to be improved in terms of clarity and results have to be tabulated and shown in tables and figures, with a clear, but concise description of results and a discussion.

Reviewers' comments:

Reviewer's Responses to Questions

**Comments to the Author**

1. Does this manuscript meet PLOS Mental Health’s publication criteria?

Reviewer #1: Yes

Reviewer #2: Partly

2. Has the statistical analysis been performed appropriately and rigorously?

Reviewer #1: Yes

Reviewer #2: N/A

3. Have the authors made all data underlying the findings in their manuscript fully available (please refer to the Data Availability Statement at the start of the manuscript PDF file)?

Reviewer #1: No

Reviewer #2: No

4. Is the manuscript presented in an intelligible fashion and written in standard English?

Reviewer #1: Yes

Reviewer #2: Yes

Reviewer #1: Manuscript Review

The present paper seeks to deepen the understanding of clients’ lived experiences with POD to inform mental health care practices that support recovery. While the manuscript addresses an important and underexplored topic, significant revisions are needed to enhance clarity and completeness before publication.

General Considerations

Although PLOS Mental Health does not impose word limits, it encourages conciseness. At approximately 10,900 words (excluding references), the manuscript resembles a thesis rather than a journal article, making it challenging to read. I recommend reducing the length to a maximum of 6,000 words—ideally around 4,000. Consider summarizing some results in tables and moving non-essential comments to an appendix. Avoid using the appendix to reiterate a model that has already been thoroughly explained in the introduction; instead, refer to existing literature where appropriate. Overall, unnecessary details should be removed from every section.

Introduction

- The transition in lines 49–50 lacks geographical context. Is this change occurring worldwide, in Europe, in Western countries, or exclusively in the Netherlands? Please clarify.

- Strengthen the discussion on the importance of clients' identity by citing:

-DOI: 10.1080/09638237.2024.2426980

-http://www.bu.edu/cpr/repository/articles/pdf/deegan1996.pdf Explicitly state that the focus is on personal recovery, distinguishing it from clinical or functional recovery.

Methods

Remove unnecessary details to streamline this section.

Results

When referring to "most" patients, specify the exact percentage where possible to enhance scientific rigor.

Summarize key results in tables to improve readability and transform the manuscript into a more accessible paper.

Discussion

A significant limitation of the study is the limited generalizability due to the small sample size. How many patients typically attend the mental health center (MHC)? Given that your study includes only a small group, provide context on its representativeness.

Reviewer #2: Authors provided a qualitative study to deepen the understanding of clients’ lived experiences with Peer-supported Open Dialogue to identify the specific behaviors, skills, and organizational elements they consider beneficial to their

recovery.

Although the topic is interesting, the manuscript’s readability is limited by the lack of tabulated data and illustrative figures. It is therefore recommended to enrich the text with tables and visual materials to improve clarity and accessibility, both in the presentation of the results and in the theoretical exposition of the POD model and its integration into the clinical workflow of MHCs.

**Do you want your identity to be public for this peer review?** For information about this choice, including consent withdrawal, please see our Privacy Policy

Reviewer #1: No

Reviewer #2: **Yes:** Michele Poletti

---

## [Decision Letter · Decision Letter 1]

4 Jan 2026

PMEN-D-25-00075R1

Transforming specialized mental health practice: insights from clients’ perspectives on how to support recovery.

PLOS Mental Health

Dear Dr. Lorenz-Artz,

Thank you for submitting your manuscript to PLOS Mental Health. After careful consideration, we feel that it has merit but does not fully meet PLOS Mental Health’s publication criteria as it currently stands. Therefore, we invite you to submit a revised version of the manuscript that addresses the points raised during the review process.

We look forward to receiving your revised manuscript.

Kind regards,

María Soledad Burrone, PhD, MPH, MD

Academic Editor

PLOS Mental Health

Journal Requirements:

Additional Editor Comments (if provided):

Reviewers' comments:

Reviewer's Responses to Questions

**Comments to the Author**

Reviewer #2: All comments have been addressed

Reviewer #3: (No Response)

publication criteria?

Reviewer #2: Yes

Reviewer #3: Yes

3. Has the statistical analysis been performed appropriately and rigorously?

Reviewer #2: N/A

Reviewer #3: N/A

4. Have the authors made all data underlying the findings in their manuscript fully available (please refer to the Data Availability Statement at the start of the manuscript PDF file)?

Reviewer #2: No

Reviewer #3: No

5. Is the manuscript presented in an intelligible fashion and written in standard English?

Reviewer #2: Yes

Reviewer #3: Yes

Reviewer #2: Authors deeply revised the paper according to editorial and reviewers' suggestion; the paper could be accepted in its current revised form

Reviewer #3: See attachment.

**Do you want your identity to be public for this peer review?** For information about this choice, including consent withdrawal, please see our Privacy Policy

Reviewer #2: **Yes:** Michele Poletti

Reviewer #3: No

---

## [Editor Report · Decision Letter 2]

3 Mar 2026

Transforming specialized mental health practice: insights from clients’ perspectives on how to support recovery.

PMEN-D-25-00075R2

Dear authors,

We are pleased to inform you that your manuscript, “Transforming specialized mental health practice: insights from clients’ perspectives on how to support recovery,” has been provisionally accepted for publication in PLOS Mental Health.

To proceed to formal acceptance, the journal office will send you a short list of formatting requirements in a follow-up email. Please note that your manuscript will not be scheduled for publication until these items have been completed, so a swift response is appreciated.

IMPORTANT: The editorial review process is now complete. At this stage, PLOS will only permit corrections to spelling, formatting, or significant scientific errors. Requests for major changes, or any that affect the scientific interpretation of the work, will delay publication.

If your institution has a press office, please notify them about your upcoming paper to help maximize its impact. If they will be preparing press materials, please inform our press team as soon as possible—no later than 48 hours after receiving the formal acceptance. Your manuscript will remain under strict press embargo until 2 pm Eastern Time on the date of publication. For more information, please contact mentalhealth@plos.org .

As a final precaution, I encourage you to review the full submission package once more to confirm (i) that any potentially identifying information has been fully anonymized (including within quotations, acknowledgments, supplementary materials, and any ethics documentation); (ii) that the Data Availability Statement is fully aligned with PLOS policy and accurately reflects any restrictions described; and (iii) that person-first, non-stigmatizing language is used consistently throughout, particularly when referring to diagnostic categories or lived experiences.

Thank you again for supporting Open Access publishing. We look forward to publishing your work in PLOS Mental Health.

Best regards,

María Soledad Burrone, PhD, MPH, MD

Academic Editor

PLOS Mental Health

Reviewer Comments:

Before final acceptance, I kindly ask that you carefully review the manuscript once more to ensure that:

All potentially identifying information has been fully anonymized, including within quotations, acknowledgments, supplementary files, and any ethics documentation.The Data Availability Statement remains fully aligned with PLOS policy and accurately reflects any restrictions described.The manuscript uses person-first and non-stigmatizing language throughout, particularly when referring to diagnostic categories or lived experiences, ensuring consistency with recovery-oriented principles.

These final checks are precautionary and intended to ensure full compliance with ethical and editorial standards prior to publication.